

# Harnessing mtDNA variation to resolve ambiguity in 'Redfish' sold in Europe

Peter Shum[1,2], Lauren Moore[2], Christophe Pampoulie[3], Cristina Di Muri[2,4], Sara Vandamme[2,5] and Stefano Mariani[2]

[1] Hopkins Marine Station, Stanford University, Pacific Grove, CA, USA
[2] School of Environment & Life Sciences, University of Salford, Salford, Greater Manchester, United Kingdom
[3] Marine and Freshwater Research Institute, Reykjavík, Iceland
[4] Evolutionary and Environmental Genomics Group, School of Environmental Sciences, University of Hull, Hull, United Kingdom
[5] North Western Waters Advisory Council, Dublin, Ireland

## ABSTRACT

Morphology-based identification of North Atlantic *Sebastes* has long been controversial and misidentification may produce misleading data, with cascading consequences that negatively affect fisheries management and seafood labelling. North Atlantic *Sebastes* comprises of four species, commonly known as 'redfish', but little is known about the number, identity and labelling accuracy of redfish species sold across Europe. We used a molecular approach to identify redfish species from 'blind' specimens to evaluate the performance of the Barcode of Life (BOLD) and Genbank databases, as well as carrying out a market product accuracy survey from retailers across Europe. The conventional BOLD approach proved ambiguous, and phylogenetic analysis based on mtDNA control region sequences provided a higher resolution for species identification. By sampling market products from four countries, we found the presence of two species of redfish (*S. norvegicus* and *S. mentella*) and one unidentified Pacific rockfish marketed in Europe. Furthermore, public databases revealed the existence of inaccurate reference sequences, likely stemming from species misidentification from previous studies, which currently hinders the efficacy of DNA methods for the identification of *Sebastes* market samples.

## INTRODUCTION

Assessing the state of marine resources often requires coordinated international effort to examine historical trends and determine changing distributions and status of fish stocks. This involves gathering catch and fishery-independent survey data, interdisciplinary evaluation on the biology and connectivity of populations (*Cadrin et al., 2010*; *Planque et al., 2013*), and monitoring of market products (*Miller, Clarke & Mariani, 2012*; *Watson et al., 2016*; *Vandamme et al., 2016*). The growing global importance of seafood trade, and the parallel advances in food technology, processing and packaging techniques, and complex supply networks make it necessary to ensure the authenticity and origin of seafood products (*Mariani et al., 2015*). Despite European Law (EC European Commission No. 1379/2013;

Corresponding author
Peter Shum, shump@stanford.edu

*European Commission, 2013*), which requests appropriate seafood traceability and labelling regulations, the identification of species is often problematic because morphologically similar species are difficult to separate by anatomical characters. As a result, resource management can be negatively affected through inflation of catch data for more desirable species and underreporting for less desirable species, and, further down the supply chain, a consequent lack of transparency for consumers.

North Atlantic (NA) *Sebastes* is comprised of four closely related species commonly referred to as 'redfish': the beaked redfish *Sebastes mentella* Travin 1951, the golden redfish *Sebastes norvegicus* (previously known as *S. marinus*) Linnaeus 1758, the Acadian redfish *Sebastes fasciatus* Storer 1854 and the Norwegian redfish *Sebastes viviparus* Krøyer 1845. They are ovoviparous (i.e., internal fertilisation), long-lived, slow-growing, late-maturing species, and have in general low natural mortality which makes them vulnerable to even low levels of harvesting (*Planque et al., 2013*). All species are commercially exploited and have experienced decreasing fishery landings since mid-1990 (*Marine Research Institute, 2014*; *Pauly & Zeller, 2015*). The identification of these species is controversial and remains difficult due to overlapping meristic and morphological features, which leads to these species being often marketed under a single vernacular name, 'redfish'. The practice of marketing species under an 'umbrella' term has important consequences because species with different conservation needs are mixed together, which, in turn, compromises the ability of consumers to make informed responsible purchasing decisions (*Griffiths et al., 2013*).

Seafood authentication and traceability has led to a recent surge in molecular-based approaches for species identification. DNA barcoding is an established technique for species identification in which DNA sequence profiles of the mitochondrial cytochrome c oxidase I (COI) for a vast range of animal species are deposited in the Barcode of Life (BOLD) database, and can subsequently be used to correctly identify unknown specimens (*Hebert, Ratnasingham & DeWaard, 2003*). There are several documented cases that illustrate the power of this approach as a tool to detect seafood mislabelling, whereby fish products can be potentially disguised for less desirable, cheaper or even illegally caught species (*Miller & Mariani, 2010*; *Miller, Clarke & Mariani, 2012*). However, the BOLD data system has a limited number of reference vouchers for some species and presents a challenging case for species that have recently diverged (*Steinke et al., 2009*). When such cases require clarification, rapidly evolving molecular markers offer additional DNA-based tools to distinguish between species (*Viñas & Tudela, 2009*; *Espiñeira & Vieites, 2012*). Mitochondrial genes are limited to only the matrilineal lineage, which can subsequently hinder the correct identification of a species, particularly when hybridisation is common like in the redfish genus (*Pampoulie & Daníelsdóttir, 2008*). However, *Shum et al. (2015)* have shown the mtDNA control region to display high resolution to distinguish *S. mentella* groups with only 4% of mismatches between mtDNA and nDNA. GenBank is a comprehensive database that contains publicly available sequences for more than 300,000 organisms, across any portion of their genomes. It locates regions of similarity between sequences, producing a list of matches most similar to the query sequence and an estimate of the percentage identity. Although less rigorously monitored, the GenBank data-base

therefore provides an additional platform to match unknown sequences to their most closely related taxa.

Here we applied a mtDNA approach to investigate the seafood trade in North Atlantic *Sebastes*, by investigating the patterns of redfish marketed from various retailers across Europe. Specifically, we sought to (1) compare the BOLD and Genbank databases to evaluate their performance in identifying blind and market specimens for the mtDNA COI and control region (d-loop) respectively, and (2) determine the identity of North Atlantic *Sebastes* species being marketed in a selection of European retailers. Results reveal that for some of these recently diverged species the classical BOLD approach remains ambiguous, and that bias in morphological identification from previous *Sebastes* studies has led to erroneous reference sequences being deposited in the GenBank database.

## MATERIALS AND METHODS

### Sample collection

We targeted North Atlantic *Sebastes* 'redfish' species from randomly chosen fishmongers or fish counters from supermarkets across Europe (Table 1). Fin clips were taken from whole bought specimens or muscle tissue from fillets and initially stored in silica before being immersed directly in 100% ethanol and stored at −20 °C. To further validate our approach to the identification of North Atlantic *Sebastes* species, we also analysed 29 blind specimens of known origin that were inspected visually (hereinafter referred to as 'MarRef'), shared by the Marine Research Institute in Iceland, and used the multimarker data set in *Shum et al. (2015)* as a reference custom data-base (Accession nos: KP988027–KP988288).

### Molecular analysis

DNA was extracted using the DNeasy kit (Qiagen, Hilden, North Rhine-Westphalia, Germany) following the manufacturer's protocol. Each sample was amplified at two markers, the universal 650 bp DNA barcode cytochrome c oxidase subunit I (COI, FishF1: 5′-TCA ACC AAC CAC AAA GAC ATT GGC AC-3′ and FishR1: 5′-TAG ACT TCT GGG TGG CCA AAG AAT CA-3′; *Ward et al., 2005*) and the mitochondrial control region (d-loop) 573 bp fragment, using Hyde and Vetter's *Sebastes*-specific primer pair (2007; D-RF: 5′-CCT GAA AAT AGG AAC CAA ATG CCA G-3′ and Thr-RF: 5′-GAG GAY AAA GCA CTT GAA TGA GC-3′). PCRs and temperature profiles for d-loop were: 94 °C (2 min), 35 cycles of (94 °C (30 s), 59 °C (60 s), 72 °C (60 s)), followed by 3 min at 72 °C; for COI: 95 °C (5 min), 35 cycles of (95 °C (30 s), 50 °C (60 s), 72 °C (60 s)) followed by 10 min at 72 °C. A negative control was included in all reactions and PCR products visualised on a 1% agarose gel. PCR products were purified and sequenced using Thr-Rf for d-loop and FishR1 for COI by SourceBioscience. Sequences were edited and trimmed using CHROMAS LITE 2.1.1 (http://technelysium.com.au/?page_id=13) and aligned with MUSCLE implemented using MEGA v.6 (*Tamura et al., 2013*).

### Data analysis

Molecular diversity indices, including nucleotide ($\pi$) (*Nei, 1987*) and haplotype ($h$) (*Nei & Tajima, 1981*) diversities, were estimated using DNASP v5.10 (*Librado & Rozas, 2009*).

**Table 1  Market sample species identification of North Atlantic *Sebastes*.** Sample code, species label information, country of origin and results from BOLD and Genbank searches using the mtDNA Cytochrome c oxidase I (COI) and Control Region (D-loop) respectively. Species separated by "/" indicate identical similarity score are considered unresolved or "ambiguous" and species in parenthesis indicated lower similarity score. Species ID is inferred from phylogenetic reconstruction based on D-loop sequences (see Fig. 2).

| Code | Sold as | Country | BOLD (COI) >99% | Genbank (D-loop) >99% | Species ID (d-loop) |
|---|---|---|---|---|---|
| Bergen_1 | redfish | Norway | *S. viviparus / S. norvegicus* | *S. fasciatus / S. marinus / S. mentella* | *S. norvegicus* |
| Bergen_2 | redfish | Norway | *S. viviparus / S. fasciatus / S. mentella* | *S. marinus / S. mentella* | *S. mentella (deep)* |
| Oslo_3 | redfish | Norway | *S. viviparus / S. fasciatus / S. mentella* | *S. mentella* | *S. mentella (shallow)* |
| Liverpool_1 | croaker | UK | *S. viviparus / S. norvegicus* | *S. norvegicus (S. fasciatus / S. mentella)* | *S. norvegicus*[a] |
| Liverpool_2 | croaker | UK | *S. viviparus / S. norvegicus* | *S. norvegicus (S. fasciatus / S. mentella)* | *S. norvegicus*[a] |
| Liverpool_3 | red snapper | UK | *S. viviparus / S. norvegicus* | *S. norvegicus (S. fasciatus / S. mentella)* | *S. norvegicus*[a] |
| Liverpool_4 | red snapper | UK | *S. viviparus / S. norvegicus* | *S. norvegicus (S. fasciatus / S. mentella)* | *S. norvegicus*[a] |
| Liverpool_5 | red snapper | UK | *S. viviparus / S. norvegicus* | *S. norvegicus (S. fasciatus / S. mentella)* | *S. norvegicus*[a] |
| Munich_1 | redfish | Germany | N/A | *S. mentella* | *S. mentella (shallow)* |
| Munich_2 | redfish | Germany | N/A | *S. mentella* | *S. mentella (shallow)* |
| Munich_3 | redfish | Germany | *S. viviparus / S. fasciatus / S. mentella* | *S. marinus (S. mentella)* | *S. mentella (deep)* |
| Munich_4 | redfish | Germany | *S. viviparus / S. fasciatus / S. mentella* | *S. marinus (S. mentella)* | *S. mentella (deep)* |
| Hamburg_1 | *S. norvegicus* | Germany | *S. viviparus / S. fasciatus / S. mentella* | *S. marinus / S. mentella* | *S. mentella (shallow)*[a] |
| Hamburg_2 | *S. alutus* | Germany | *S. polyspinis (S. crameri / S. reedi)* | Unidentified | Unidentified |
| Hamburg_3 | *S. norvegicus* | Germany | *S. viviparus / S. norvegicus* | *S. norvegicus (S. fasciatus / S. mentella)* | *S. norvegicus* |
| Hamburg_4 | *Sebastes sp.* | Germany | *S. viviparus / S. norvegicus* | *S. norvegicus (S. fasciatus / S. mentella)* | *S. norvegicus* |
| Leuven_1 | *S. norvegicus* | Belgium | *S. viviparus / S. norvegicus* | *S. norvegicus (S. fasciatus / S. mentella)* | *S. norvegicus* |
| Leuven_2.1 | *S. norvegicus* | Belgium | *S. viviparus / S. norvegicus* | *S. norvegicus (S. fasciatus / S. mentella)* | *S. norvegicus* |
| Leuven_2.2 | *S. norvegicus* | Belgium | *S. viviparus / S. norvegicus* | *S. norvegicus (S. fasciatus / S. mentella)* | *S. norvegicus* |

Notes.

[a]Mislabelled samples.

N/A, no available sequence.

*S. marinus* = *S. norvegicus*.

The COI sequences were identified using the Barcode of Life Data Systems online (BOLD, Biodiversity Institute of Ontario, University of Guelph, Guelph, Ontario, Canada; http://www.barcodinglife.org; *Ratnasingham & Hebert, 2007*) using the species-level barcode database to identify each sequence. The d-loop sequences were cross-referenced using BLAST on GenBank (Basic Local Alignment Search Tool, National Centre for Biotechnology Information, Bethesda, Maryland; http://www.ncbi.nlm.nih.gov/). A threshold of 99% to 100% sequence similarities was used above which identification of unknown samples was deemed reliable.

## Sequence analysis

Blind and market samples were analysed along with a subset of seven previously characterised samples of North Atlantic *Sebastes*, which included *S. viviparus*, *S. fasciatus*, *S. norvegicus* and shallow and deep *S. mentella* types (*Shum et al., 2015*; see Supplemental Information for Genbank accessions numbers). Sequence identification was strengthened via phylogenetic analysis, nesting blind and market samples together with a subset of reference samples from *Shum et al. (2015)*. Furthermore, all publicly available published sequence data of North Atlantic *Sebastes* was included from Genbank as a further source

of reference that derived from two more studies. First, we obtained COI and d-loop data generated by *Hyde & Vetter (2007)* for all four North Atlantic *Sebastes*, including selected Pacific species (see further details are illustrated in Supplemental Information), which was used to investigate the evolutionary relationships of over 100 *Sebastes* species. Second, we obtained d-loop data generated from *Artamonova et al. (2013)* that was used to investigate hybridisation and diversification of redfish in the Irminger Sea. Phylogenetic relatedness among sequences was reconstructed in MEGA version 6 (*Tamura et al., 2013*) using the neighbor-joining (NJ) algorithm (*Saitou & Nei, 1987*), and the Kimura 2-parameter distance. The NJ approach was used as it is a simple approach to determine the position of *Sebastes* 'products' within 'species clusters'. All positions containing alignment gaps and missing data were eliminated only in pairwise sequence comparisons (Pairwise deletion option). Evaluation of statistical confidence in nodes was based on 1,000 non-parametric bootstrap replicates (*Felsenstein, 1985*).

## RESULTS

### Marker diversity

A total of 48 samples were analysed of which 19 were collected from 12 different retailers in four European countries (Belgium, Germany, Norway and the UK) and the remaining 29 blind samples were obtained from the Marine Research Institute in Iceland. The comparison of interspecific genetic variability for all blind and market samples considered recovered a higher number of haplotypes ($h$) from the mtDNA d-loop ($h = 16$, $h \pm SD = 0.863 \pm 0.034$; $\pi \pm SD = 0.018 \pm 0.003$) with a 3-fold greater nucleotide diversity ($\pi$) than the COI ($h = 11$, $h \pm SD = 0.838 \pm 0.035$; $\pi \pm SD = 0.006 \pm 0.001$) marker.

### Sequence identification

Sequences were initially screened using the DNA sequence search databases BOLD and BLAST to identify the closest matching sequences for COI (420–651 bp, average length 544 bp) and d-loop (381–506 bp, average length 486 bp) respectively. The BOLD search for market and blind samples resolved only 31% ($N = 12/39$) of the matches allowing for assignment at 99% or 100% where no other species possessed the same similarity score, while 69% ($N = 27/39$) were unresolved or "ambiguous" due to high sequence similarity with more than one redfish species or could not be reliably identified at species level (e.g., *Sebastes* spp.). On the other hand, the BLAST search of *Sebastes* d-loop sequences returned 92% ($N = 44$) of the matches identified at 100% or 99% where no other species possessed the same similarity score, while 8% ($N = 4$) of the matches produced ambiguous results (Tables 1 and 2).

### Phylogenetic analysis

Phylogenetic reconstruction under a neighbor-joining framework was conducted with reference haplotypes for each species, *S. viviparus*, *S. fasciatus*, *S. norvegicus*, and shallow- and deep-pelagic *S. mentella*. Included in the phylogenetic analysis were 29 blind specimens and 19 market samples. Selected Pacific *Sebastes* sequences were included in subsequent analysis because one product was sold as *S. alutus* and therefore set *S. aleutianus* as outgroup

**Table 2  Blind sample species identification of North Atlantic *Sebastes*.** Samples code and results from BOLD and Genbank searches using the mtDNA Cytochrome c oxidase I (COI) and control region (D-loop) respectively. Species separated by "/" indicate identical similarity score are considered unresolved or "ambiguous" species identification and species in parenthesis indicated lower similarity score. Species ID is inferred from phylogenetic reconstruction based on D-loop sequences (see Fig. 2). Morphological classification refers to original visual morphological identification (MarRef).

| Code | BOLD (COI) >99% | Genbank (D-loop) >99% | Morphological classification | Species ID (D-loop) |
|---|---|---|---|---|
| B_1 | N/A | *S. mentella* | *S. mentella* Icelandic shelf west | *S. mentella* (shallow)[a] |
| B_2 | *S. viviparus* / *S. norvegicus* | *S. norvegicus* (*S. fasciatus* / *S. mentella*) | Correct | *S. norvegicus* |
| B_3 | *S. viviparus* (*S. fasciatus* / *S. mentella*) | *S. viviparus* | Correct | *S. viviparus* |
| B_4 | *S. viviparus* (*S. fasciatus* / *S. mentella*) | *S. viviparus* | Correct | *S. viviparus* |
| B_5 | *S. viviparus* / *S. norvegicus* | *S. norvegicus* (*S. fasciatus* / *S. mentella*) | Correct | *S. norvegicus* |
| B_6 | N/A | *S. mentella* | *S. mentella* (shallow) | *S. mentella* (deep)[a] |
| B_7 | N/A | *S. mentella* | *S. mentella* Icelandic shelf west | *S. mentella* (shallow)[a] |
| B_8 | *S. viviparus* (*S. fasciatus* / *S. mentella*) | *S. viviparus* | Correct | *S. viviparus* |
| B_9 | N/A | *S. norvegicus* (*S. fasciatus* / *S. mentella*) | Correct | *S. norvegicus* |
| B_10 | *S. viviparus* / *S. fasciatus* / *S. mentella* | *S. mentella* | Correct | *S. mentella* (deep) |
| B_11 | *S. viviparus* / *S. fasciatus* / *S. mentella* | *S. mentella* | Correct | *S. mentella* (shallow) |
| B_12 | *S. viviparus* / *S. fasciatus* / *S. mentella* | *S. mentella* | *S. mentella* Icelandic shelf west | *S. mentella* (shallow)[a] |
| B_13 | *S. viviparus* (*S. fasciatus* / *S. mentella*) | *S. viviparus* | Correct | *S. viviparus* |
| B_14 | *S. viviparus* / *S. fasciatus* / *S. mentella* | *S. mentella* | Correct | *S. mentella* (shallow) |
| B_15 | *S. viviparus* (*S. fasciatus* / *S. mentella*) | *S. viviparus* | Correct | *S. viviparus* |
| B_16 | N/A | *S. mentella* | *S. mentella* (deep) | *S. mentella* (shallow)[a] |
| B_17 | N/A | *S. mentella* | Correct | *S. mentella* (shallow) |
| B_18 | *S. mentella* (*S. viviparus* / *S. fasciatus*) | *S. mentella* | Correct | *S. mentella* (deep) |
| B_19 | *S. mentella* (*S. viviparus* / *S. fasciatus*) | *S. norvegicus* (*S. fasciatus* / *S. mentella*) | *S. mentella* (shallow) | *S. norvegicus*[a] |
| B_20 | *S. viviparus* / *S. fasciatus* / *S. mentella* | *S. marinus* (*S. mentella*) | *S. norvegicus* | *S. mentella* (deep) |
| B_21 | *S. mentella* (*S. viviparus* / *S. fasciatus*) | *S. mentella* | Correct | *S. mentella* (deep) |
| B_22 | *S. mentella* (*S. viviparus* / *S. fasciatus*) | *S. mentella* | Correct | *S. mentella* (deep) |
| B_23 | *S. mentella* (*S. viviparus* / *S. fasciatus*) | *S. mentella* | *S. mentella* (shallow) | *S. mentella* (deep)[a] |
| B_24 | *S. viviparus* / *S. fasciatus* / *S. mentella* | *S. mentella* | *S. mentella* (deep) | *S. mentella* (shallow)[a] |
| B_25 | *S. viviparus* / *S. fasciatus* / *S. mentella* | *S. mentella* | Correct | *S. mentella* (shallow) |
| B_26 | *S. mentella* (*S. viviparus* / *S. fasciatus*) | *S. mentella* | Correct | *S. mentella* (deep) |
| B_27 | *S. viviparus* / *S. fasciatus* / *S. mentella* | *S. mentella* | Correct | *S. mentella* (shallow) |
| B_28 | *S. mentella* (*S. viviparus* / *S. fasciatus*) | *S. mentella* | *S. mentella* (deep) | *S. mentella* (shallow)[a] |
| B_29 | N/A | *S. mentella* | Correct | *S. mentella* (deep) |

**Notes.**
[a] Misidentified samples.
N/A, no available sequence.
*S. marinus* = *S. norvegicus*.

(Figs. 1 and 2; see *Hyde & Vetter, 2007*). The two mitochondrial markers show discordant phylogenetic patterns. The COI topology was ambiguous since *S. fasciatus* and *S. mentella* appear to be paraphyletic, while the d-loop topology was consistent with the specific definition of each taxon, strongly recovering North Atlantic species in independent clades, including shallow- and deep-pelagic *S. mentella* (Figs. 1 and 2). Thus, the criteria used to assess species/population assignment of blind and market specimens were based on inferences from the d-loop phylogeny. Identification of blind specimens inferred from

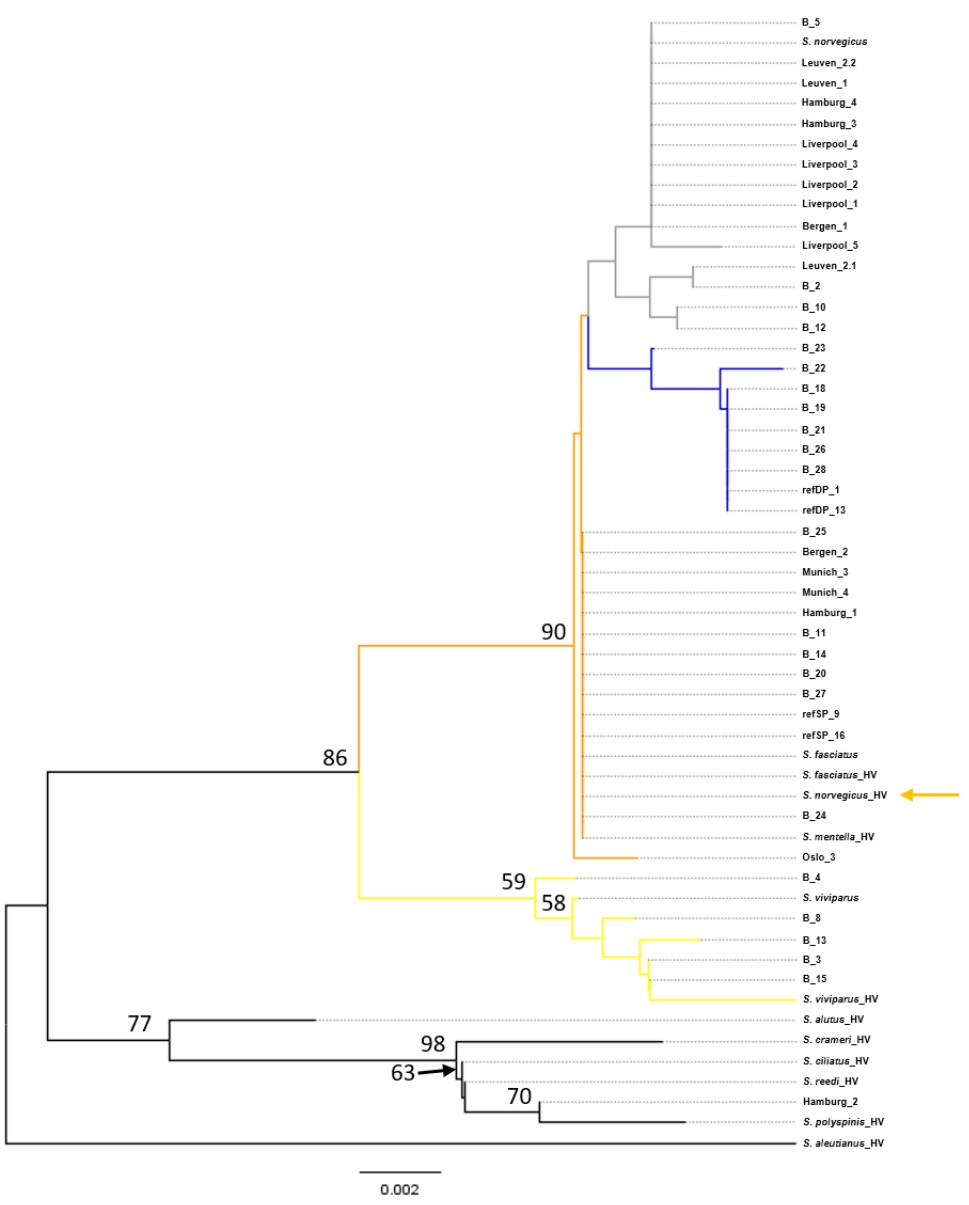

**Figure 1** **Neighbor-Joining tree of mtDNA COI sequences.** Phylogenetic tree using the seven generated reference sequences representing the four recognized redfish species and *S. mentella* shallow- and deep-types and including COI sequences of 10 species from *Hyde & Vetter (2007)*. Numbers at the nodes represent bootstrap support, with <50% absent after 1,000 replicates. HV indicates *Hyde & Vetter*'s (*2007*) generated sequences and orange arrow shows erroneous reference entry. Reference entries (this study): *S. viviparus*, *S. fasciatus*, *S. norvegicus* shallow-pelagic *S. mentella*: refSP, deep-pelagic *S. mentella*: refDP. Colour of the branches represent species/population assignment, yellow: *S. viviparus*; gray: *S. norvegicus*; orange: *S. fasciatus/S. mentella* (shallow-pelagic); blue: *S. mentella* (deep-pelagic).

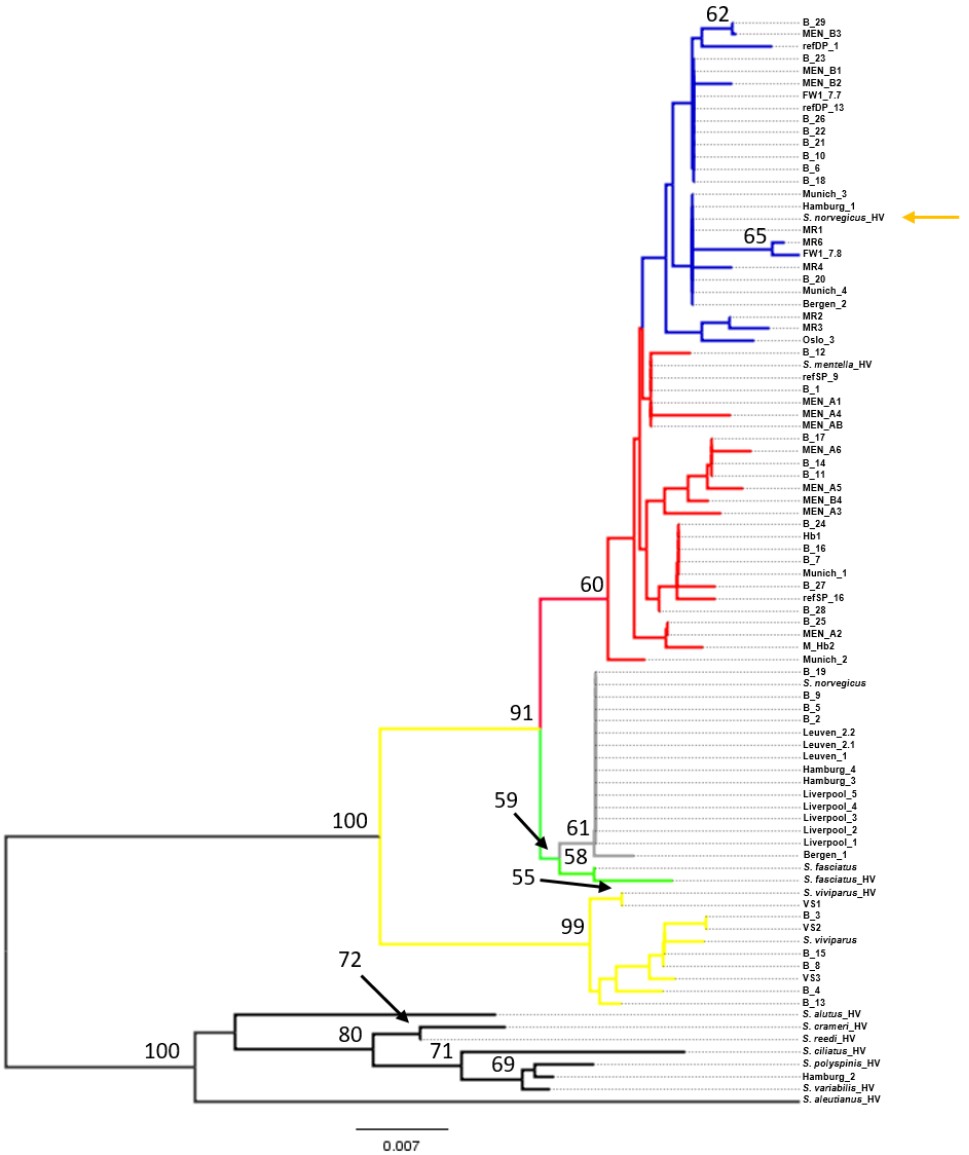

**Figure 2 Neighbor-Joining tree of mtDNA control region sequences.** Phylogenetic tree using the seven generated reference sequences representing the four recognized redfish species and *S. mentella* shallow-and deep-types and including control region sequences from eleven Pacific species from *Hyde & Vetter (2007)*. Numbers at the nodes represent bootstrap support, with <50% absent after 1,000 replicates. HV indicates *Hyde & Vetter*'s (*2007*) sequences and orange arrow shows erroneous reference entry. Reference entries (this study): *S. viviparus*, *S. fasciatus*, *S. norvegicus* shallow-pelagic *S. mentella*: refSP, deep-pelagic *S. mentella*: refDP. Colour of the branches represent species/population assignment, yellow: *S. viviparus*; green: S. fasciatus; gray: *S. norvegicus*; red: *S. mentella* (shallow-pelagic); blue: *S. mentella* (deep-pelagic). FW: *S. mentella* (deep–Faroe west). From (*Artamonova et al., 2013*); VS: *S. viviparus*; MEN: *S. mentella*; M_hb2 & hb1: *S. mentellaS.viviparus* hybrid; MR: *S. norvegicus* (which group with *S. mentella* (deep-pelagic) haplotypes).

the d-loop phylogeny was cross-referenced against records from the Marine Research Institute - MarRef. Assessment of blind test samples were generally in agreement with the exception of samples 'B_19 and B_20' which cluster with *S. norvegicus* and *S. mentella* (deep) despite being visually classified as *S. mentella* (shallow) and *S. norvegicus* respectively (Table 2, Fig. 2). Furthermore, five mismatches were identified concerning the shallow- and deep-pelagic *S. mentella* types. Two specimens (B_6 and B_23) cluster with the deep-pelagic group, while being previously identified as belonging to the shallow-pelagic group. Similarly, three specimens (B_16, B_24 and B_28) cluster with the shallow-pelagic group but were originally identified to belong to the deep-pelagic group.

For the market label analysis, *S. norvegicus* and *S. mentella* (shallow- and deep-pelagic types) were the most commonly available species in markets. Products from Belgium were labelled as *S. norvegicus* and nest with the *S. norvegicus* references. German products from Munich sold as 'Rotbarsch' (*Sebastes* spp.) clustered with shallow-pelagic (Munich_1 & Munich_2) and deep-pelagic (Munich_3 & Munich_4) *S. mentella*. In Hamburg, however, one sample labelled '*Sebastes* spp.' grouped with *S. norvegicus* (Hamburg_4) while two products sold as '*S. norvegicus*' and '*S. alutus*' group with shallow-pelagic *S. mentella* (Hamburg_1) and an unidentified Pacific *Sebastes* species respectively; although the sample sold as '*S. alutus*' could not be reliably resolved at species level (Fig. 2). Norwegian products sold as 'redfisk' (redfish) grouped with *S. norvegicus* (Bergen_1) and deep-pelagic *S. mentella* (Bergen_2), while one product from Oslo grouped with shallow-pelagic *S. mentella* (Oslo_3). Among the UK specimens, five samples sold as 'croaker' ($N = 2$, Liverpool_1 & Liverpool_2) and 'snapper' ($N = 3$, Liverpool_3-L Liverpool_5) cluster with *S. norvegicus*.

## DISCUSSION

North Atlantic (NA) *Sebastes* is represented by four species (*S. viviparus*, *S. fasciatus*, *S. norvegicus* and *S. mentella*) that can be sold under a common market name, 'redfish', which does not have a uniform designation of species in different European countries (*Ministry of Industry and Ministry of Fisheries, 2013*; *Federal Agency for Agriculture and Food, 2015*). The practice of marketing multiple species under one vernacular name hinders the identification of sea-food products and, consequently, has important implications for consumer choice and conservation (*Griffiths et al., 2013*). Thus, DNA based methods are often required to authenticate the correct labelling of fish, particularly when morphological traits are removed. Here we show that the perception of much of the biocomplexity of *Sebastes* is compromised using the 'redfish' umbrella term. We also find that, irrespective of the market name used, the universal COI approach to identify species is inadequate for this group. The use of a more variable mtDNA fragment (d-loop) allows for the distinction of monophyletic groups, including the four species and the two *S. mentella* 'shallow' and 'deep' types (*Shum et al., 2015*). However, the use of public databases is currently compromised by wrong reference sequence entries.

## Complicating issues for DNA barcoding for recently radiating species

North Atlantic redfish species have a recent evolutionary history, having diversified during the Pleistocene (*Hyde & Vetter, 2007*; *Shum et al., 2015*) and exhibit overlapping meristic and morphological characteristics. This makes phenotypic-based species identification difficult between these closely related species, which may result in misclassification of individuals (*Pampoulie & Daníelsdóttir, 2008*). Many studies have shown the effectiveness of DNA barcoding using the mitochondrial cytochrome c oxidase I (COI) gene for species identification in a wide range of animal species (*Wong & Hanner, 2008*; *Filonzi et al., 2010*). However, universal COI barcoding presents a challenge to discriminate between recently diverged species, due to the rather conserved COI gene among congeneric species (*Steinke et al., 2009*; *Viñas & Tudela, 2009*). This study exemplifies this issue in the context of *Sebastes* DNA-based identification.

Firstly, the BOLD repository contains a limited availability of representative voucher specimens and reference sequences. This is particularly the case for North Atlantic redfish as the BOLD search of the COI barcodes identified multiple species that fall within the 2% divergence threshold. These matches included similar identity scores between *S. norvegicus* and *S. viviparus* and between *S. mentella* and *S. norvegicus*. Although the pool of blind specimens did not include *S. fasciatus*, a BOLD query of the reference samples could not be identified at species level (i.e., *Sebastes* sp.). Similarly, the BOLD database failed to identify eight of the blind specimens to species level that were classified to be *S. mentella*. Thus, DNA barcoding has poor resolution for this young group due to insufficient reference sequences and/or lacks the diagnostic polymorphisms to accurately identify them.

The mitochondrial control region (d-loop), however, is a fast-evolving marker and has been reported to reliably distinguish between North Atlantic redfish species (*Shum et al., 2015*). D-loop sequences queried against the Genbank database produced a higher number of positive and unambiguous matches and resolved all the unidentified specimens that the COI failed to classify (e.g., Oslo_3, Munich_3, Munich_4, Hamburg_1, B_10, B_11, B_12, B_14, B_20, B_24, B_25 & B_27; Table 1). Genbank is a larger database than BOLD and contains more sequences that will subsequently increase the identification of unknown samples. However, it contains a mixture of verified and unverified sequences without appropriate quality-control procedures, which may populate the database with ambiguous sequence submissions (*Wong & Hanner, 2008*). Similar to COI, a BLAST search of the d-loop sequences identified multiple species that fall within the 2% divergence threshold, notably between *S. norvegicus* (100%), *S. fasciatus* (99%) and *S. mentella* (99%) (Table 1). Sequences uploaded into Genbank are potentially erroneous due to morphological mis-identification, however, *Shum et al. (2015)* provide a comprehensive dataset of all North Atlantic *Sebastes* (d-loop) in Genbank, which can be used as a reliable reference (Accession nos: KP988027–KP988288).

While the d-loop exhibits the resolution to unambiguously distinguish among the four species and the two main lineages within *S. mentella* (Fig. 2), it should be highlighted that the use of mitochondrial genes is limited because only the matrilineal lineage is examined, which could hinder the correct identification of a species, in hybridizing groups.

*Pampoulie & Daníelsdóttir (2008)* reported evidence of introgressive hybridization among North Atlantic redfish and discovered individual misclassification based on morphological identification. Consequently, hybrid redfish will be assigned to the maternal parent, which can further complicate the application of molecular-based identification approaches (*Nicolè et al., 2012*).

## Phylogenetic representation

The phylogenetic approach was based on the construction of a Neighbor-joining (NJ) tree to assess the results of the database searches (Figs. 1 and 2). The NJ reconstruction of the COI marker could distinguish between *S. viviparus*, *S. norvegicus* and the deep-pelagic *S. mentella*, but was not diagnostic between *S. fasciatus* and shallow-pelagic *S. mentella*. However, the phylogenetic reconstruction of the mtDNA control region allowed full discrimination of all species into distinct monophyletic groups, including the shallow-pelagic and deep-pelagic *S. mentella* (Fig. 2). *Shum et al. (2015)* have examined the extent of population structure in *S. mentella* in the North Atlantic and show the existence of two strongly divergent groups—a shallow-pelagic population showing a homogenous distribution throughout the North Atlantic above 500 m and a deep-pelagic found to the central North Atlantic below 500 m. It is suggested that climactic and oceanographic processes may have shaped the current *S. mentella* population structure during the Pleistocene glaciations (*Shum et al., 2015*). Furthermore, *Pampoulie & Daníelsdóttir (2008)* reported significant levels of hybridization suggesting the shallow- and deep-pelagic groups were allopatric before secondarily coming into contact to form their current sympatric distribution (*Cadrin et al., 2010*).

The assessment of the blind specimens revealed unambiguous clustering of species with the exception of two misclassifications (B19 & B20). These specimens were morphologically classified visually but were identified as another species using the mtDNA control region (Table 2). This, perhaps, is indicative of misidentification or potentially hybridisation which could be further investigated using nuclear markers (*Pampoulie & Daníelsdóttir, 2008*). Moreover, we found discrepancies concerning the visual identification of the shallow- vs. deep-type *S. mentella*. These two types are often separated by their morphological appearance (*Stefánsson et al., 2009*), their depth and geographic distribution, yet 17% (5 of 29) of the blind specimens were inaccurately identified. These patterns may have cascading consequences for conservation management and seafood labelling.

The market-level analysis exemplifies patterns of misidentification for NA redfish. Among the market collections, two North Atlantic (*S. norvegicus* and *S. mentella* shallow- and deep-types) and one ambiguous North Pacific species were recorded. Redfish collected in the UK were marketed under various, inaccurate labels, including 'croaker' and 'red snapper', all of which were identified as *S. norvegicus*. This reveals a lack of knowledge, raising serious concern about the redfish trade in the UK. In Europe, it is required that fish product labels contain the scientific name as well as production method and area where the product was caught (EC No 1379/2013). However, in some cases, German fish mongers or companies write 'Rotbarsch or *Sebastes* spp.' because the German index of designation names for fish species allows species in the genus *Sebastes* spp., not identified to species

level, to be called 'Rotbarsch' (*Federal Agency for Agriculture and Food, 2015*). Therefore, products from Munich and one product from Hamburg fall under the umbrella term 'Rotbarsch' that comprised of shallow- and deep-pelagic *S. mentella* and *S. norvegicus*, all of which have different conservation needs (*Marine Research Institute, 2014*). While the majority of the labelled redfish products were grouped with the species listed on the label, there were instances of mislabelling. Two samples, Hamburg_1 and Hamburg_2, exhibit mitochondrial sequences that did not group with the designated label product indicating that these products are mislabelled. In Norway, the designation name for fish species sold as 'redfisk' can only refer to *S. marinus* (*S. norvegicus*), while *S. mentella* must be correctly labelled *S. mentella* and sold under the common name 'rosefish' (*Ministry of Industry and Ministry of Fisheries, 2013*). Therefore, incidence of mislabelling was recovered from Bergen ($N = 1$) and Oslo ($N = 1$) as these products were labelled 'redfisk' but identified as *S. mentella*.

An important issue revealed in the present study concerned the sale of North Atlantic *Sebastes* under a single vernacular name, redfish. In Europe, the umbrella term 'redfish' shows conflicting definitions among nations and thus, creates confusion in the redfish industry regarding which species are targeted under this term. The practice of selling multiple species under a common name has important consequences for market driven conservation strategies and prevents consumers from making informed decisions about sustainable purchasing (*Logan et al., 2008*). Furthermore, in the United States, *Logan et al. (2008)* identified five Pacific *Sebastes* (rockfish) species sold as "Pacific Red Snapper" in outlets across California and Washington. In the past, thirteen rockfish species could be sold under this common vernacular name before new regulation state that "Pacific Red Snapper" could no longer be an acceptable market name for rockfish species (*Willette et al., 2017*). This issue presented a challenge in the identification of *Sebastes* species, both Atlantic and Pacific species, because a recent diversification of the group (*Hyde & Vetter, 2007*) and the occurrence of introgressive hybridization between close relatives (*Roques, Sevigny & Bernatchez, 2001*; *Pampoulie & Daníelsdóttir, 2008*) led to ambiguous molecular taxonomy using classical COI barcoding markers (*Steinke et al., 2009*). For example, two 'redfish' samples collected from one retailer in Bergen, Norway, clustered as two different species, *S. norvegicus* and *S. mentella*, which have markedly different life history characteristics and conservation needs (Table 1; *Cadrin et al., 2010*; *Marine Research Institute, 2014*). Moreover, *S. mentella* possess at least two genetically distinguishable groups with different management advice, with a moratorium placed on the shallow population and the deep population limited to a catch limit of 20,000 tonnes/year (*Cadrin et al., 2010*; *Shum et al., 2015*). Yet, we identified both *S. mentella* types in Germany under the common name, Rotbarsch (redfish), which eliminates the ability of consumers to choose less vulnerable groups.

## Resolution of species-population identity

An important pattern revealed in the present study concerns the inclusion of misclassified redfish species in genetic databases, which may lead to erroneous conclusions when querying sequence repositories. *Hyde & Vetter (2007)* examined the evolutionary

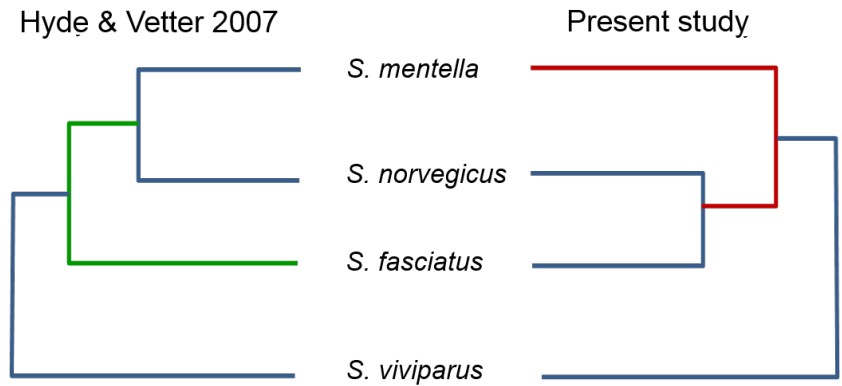

**Figure 3  Proposed North Atlantic *Sebastes* phylogenies of *Hyde & Vetter (2007)* compared to the present study.**

relationships of 101 *Sebastes* species, within which the four Atlantic species occupy a relatively peripheral position. However, by comparing the topology of the Atlantic clade in their study and the present work, using the same d-loop fragment, we noticed that the reciprocal relationships among *S. mentella*, *S. norvegicus* and *S. fasciatus* were different (see Fig. 3), with *S. norvegicus* appearing closer to *S. mentella* in *Hyde & Vetter (2007)*, while clustering as sister taxon of *S. fasciatus* in our tree, in line with *Shum et al. (2015)*.

Although the topology could be affected by different numbers of sequences compared, we included the sequence data from *Hyde & Vetter (2007)* and discovered that their *S. norvegicus* sample clustered within the deep-type *S. mentella* (Fig. 2), indicating that their reference material was a result of a misidentified deep-type *S. mentella* or a *S. norvegicus* X *S. mentella* deep-type hybrid. The distinction between *S. mentella* and *S. norvegicus* has been reported to be extremely difficult using morphological traits (*Power & Ni, 1985*; *Rubec et al., 1991*) and it is expected that misclassification can occur. An even more notable example is offered by the comparison with *Artamonova et al. (2013)*, who evaluated hybridization of *S. mentella* and reportedly included representative NA redfish species in their study 'which could be identified with certainty by morphology studies and allozyme analysis' (pp. 1794, paragraph 12 in *Artamonova et al., 2013*). Strikingly, six of the putative *S. norvegicus* haplotypes representing 15 individuals (collected at Bear Island Trough & Kopytov area, northeast of the Norwegian Sea) all consistently cluster within the shallow- and deep-type *S. mentella* (Fig. 2). Their *S. norvegicus* haplotypes mostly cluster with the deep-pelagic group from reference samples collected west of the Faroes. Moreover, they seem to maintain integrity from the deep-pelagic group found in the Irminger Sea and west of the Faroes, indicating further population structuring than previously thought (See Fig. 4, *Shum et al., 2015*). This example stresses the inherent difficulty in discriminating phenotypes and hence, interpreting results for North Atlantic *Sebastes*. Thus, it is important that publicly available databases are carefully checked and verified, especially for species that are both 'difficult' and important, so that the power of DNA identification can be properly harnessed.

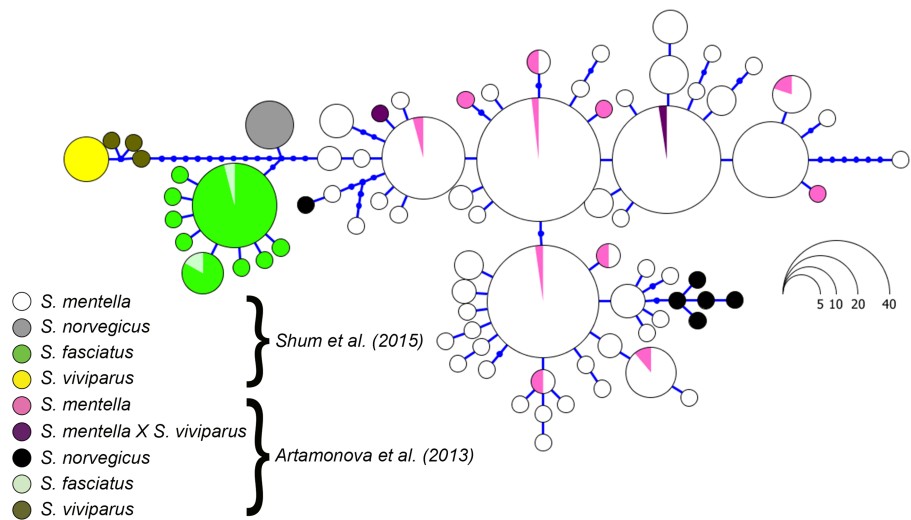

**Figure 4** **Haplotype genealogy of North Atlantic *Sebastes* from *Shum et al. (2015)*.** Data from *Artamonova et al. (2013)* is visualised among reference haplotypes where their *S. norvegicus* haplotypes cluster with shallow- and deep-type *S. mentella*.

## CONCLUSIONS

Our study focused on the molecular identification of North Atlantic *Sebastes* to clarify the status of this young group and present the first attempt to investigate the patterns of sale of North Atlantic *Sebastes* across Europe. We confirmed the inadequacy of the COI universal barcode to discriminate among species within this commercially important genus. We also illustrate the difficulty of publicly available databases to provide reliable matches of unknown specimens caused by scarce or erroneous reference entries. However, the use of the d-loop locus in a phylogenetic framework provided a higher resolution in identifying unknown specimens to species/population level of North Atlantic *Sebastes*. We report examples whereby morphological misidentification may have led to misleading data. Monitoring of accuracy in public databases and the introduction of sensitive nuclear markers therefore represent the next necessary steps to gain a fuller understanding of the biological complexity of this important commercial resource while ensuring its market traceability.

## ACKNOWLEDGEMENTS

We thank Adriana Adolfi for assistance in sample collection in Norway, Ilaria Coscia & Ute Schröder for samples collected from Leuven, Belgium and Hamburg, Germany respectively. We are grateful to Jens Carlsson & Jean Boubli for feedback on earlier versions of the manuscript.

### Funding

This work was funded by the Marine and Freshwater Research Institute in Iceland and the University of Salford GTS/PtE programme. The funders had no role in study design, data collection and analysis, decision to publish, or preparation of the manuscript.

### Grant Disclosures

The following grant information was disclosed by the authors:
Marine and Freshwater Research Institute in Iceland.
University of Salford GTS/PtE programme.

### Competing Interests

The authors declare there are no competing interests.

### Author Contributions

- Peter Shum conceived and designed the experiments, performed the experiments, analyzed the data, contributed reagents/materials/analysis tools, wrote the paper, prepared figures and/or tables, reviewed drafts of the paper.
- Lauren Moore, Cristina Di Muri and Sara Vandamme performed the experiments, reviewed drafts of the paper.
- Christophe Pampoulie analyzed the data, contributed reagents/materials/analysis tools, reviewed drafts of the paper.
- Stefano Mariani conceived and designed the experiments, contributed reagents/materials/analysis tools, wrote the paper, reviewed drafts of the paper.

### Animal Ethics

The following information was supplied relating to ethical approvals (i.e., approving body and any reference numbers):

The College of Science and Technology Research Ethics Panel (CST) provided full approval for this research involving vertebrate animals. REP Reference: CST 13/107.

### DNA Deposition

The following information was supplied regarding the deposition of DNA sequences:
Genbank MF352036–MF352136.

### Data Availability

Sequences have been submitted to Genbank accession numbers MF352036–MF352136 and raw data is presented in the Supplemental Information provided.

### Supplemental Information

Supplemental information for this article can be found online at http://dx.doi.org/10.7717/peerj.3746#supplemental-information.

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
