# Peer review of "Harnessing mtDNA variation to resolve ambiguity in ‘Redfish’ sold in Europe"

_PeerJ, doi:10.7717/peerj.3746_

## Round 0.1 · original submission · Minor Revisions

The reviewers raise a number of minor questions, points and corrects that I believe are relatively simple to address.

Reviewer 1 ·

Basic reporting

The scope is interesting sine the adequacy or not of COI universal code must be taken into account. A suitable genetic marker selection is a big concern in traceability issues, to assure the transparency across the complete food chain, and sustainibility.

Table 1 & 2 are very confused

In general the presenting of results is confused, including Fig 1 &2

Experimental design

Materials and Methods

Sequence analysis:

-line 135; subset of previously samples...which included (Shum et al, 2015): how many?
-line 143; see results?? in Mat&Meth section?? what do you mean?

In the analysis a number of previously published sequences have been included in the analysis. Include them in a table includeingthe reference etc...

Results:

line 160; Clarify and specify the meaning of "h" and "π" data
line 168-174; Percentages are confused:
30% I guess 12/41
70% neither 27/41 nor 27/48 how do you calculate?
92% I guess 44/48
8% I guess 4/48
Please make easier the presentation of results. Improve Table 1 & 2 defining "ambigous results" in order to make understandable. Is very diffiult to identify the mislabeled samples
line 170; Define what is for authors one "ambiguous results"

In the species id you conclude that an unknow sample is shallow or deep (S. mentella). However, you conclude this with only 6 reference samples (Table 2) In my iponion you can not resolve this aspects with this reduced number of samples

Sequence analysis (Fig 1&2):

You can not included in the building of a NJ phylogenetic tree all samples including, in the same analysis, "market" and "blind/references" sequences. This fact produces a very important and significant bias. First, you must build a tree only with the reference samples and after doing this analysis you can include the unknown samples but "one by one" to resolve with accuracy the species i.d.

In Fig 2, you say that red group is S. mentella (shallow). Attending to Table 2 classification; B6, B19 and B23 are S. mentella (shallow) but none of these reference samples are aligned in the red branches, can you explain it?, I'm confused


Discussion:

line 216; (see below) where? be more specific
line 285; "visual eye", do you mean "attending to taxonomic keys"?? Please modify it
line 298; "alarming discovery"" This statement is to strong. You have analysed only 5 samples from UK!!!! You demonstrate nothing with this small sampling size

Validity of the findings

Some statements are very ambitious and are not demonstrated with the findings

Additional comments

References can be improved. I miss a number of important references concerning to authentication of seafood products and the use of genetic markers.

·

Basic reporting

This paper takes focused look at the identification and labeling of commercially exploited North Atlantic Sebastes species. It addresses the difficulties of genetic and morphological identification in a recently diverged group of fishery species, proposes an alternative marker (d-loop) to the COI standard in BOLD, and examines labeling accuracy in northern Europe. Overall, the paper is well-written and well-organized, and the figures and tables are good. I have a few minor grammatical changes (suggested below), and a few larger suggestions to expand the methods and discussion (suggested in appropriate subheadings). However, all of these are fairly minor, and with some revision I believe this manuscript meets PeerJ’s standards and would be a solid addition to the literature.

Specific comments:
I found it surprising that the authors made little reference to the issues inherent in Pacific Sebastes identification. There are very similar complications in this group, particularly with regard to vague commercial labeling of multiple species. A brief comparison between the Atlantic and Pacific groups would be a nice addition to the discussion, and help to broaden its context. The authors already reference a paper on this (Logan et al. 2008); it would be nice to see it discussed in a little more detail in the discussion.

Experimental design

The experimental design was generally fine. The authors did a nice job of laying out the difficulties and importance of identifying north Atlantic Sebastes species, and their sampling was good. Their inclusion of blind genotyping of 29 samples morphologically identified by outside experts was a particularly strong element of the study.

Validity of the findings

The conclusions were generally well-supported and the authors addressed the study’s limitations fairly. In particular, they clearly acknowledged the potential for hybridization to confound any approach based on mitochondrial markers.

Specific comments:
How many sequences listed as your four target species were present in GenBank and BOLD for d-loop / COI at the time of your search? I would like a bit more reassurance that the difference in identification between the databases wasn’t confounded by very different reference availability. Likewise, I would like to see a GenBank search for COI; this would allow more direct comparison between the databases. I do agree that the phylogenetics presented later suggest that d-loop is just a better marker for this group. However, since the paper directly compares the two databases, it would be helpful to have a more direct comparison and a brief discussion of the impact of sequence availability on database-specific results.

Deep-vs-shallow S. mentella: Is there really such a bright line between these groups? Your d-loop tree suggests a lot of diversity. Would like to see a little more discussion of what might be causing the ambiguity – eg, since it is classified as an “incipient” divide, is there potential ongoing introgression between “deep” and “shallow” S. mentella beyond occasional hybridization?

Additional comments

Figures 1 & 2: This is kind of a chronic issue with large trees, but it is pretty difficult to make out/interpret the sample names, even printed full-page. Would really like to see something done to improve the size and clarity of the names. More context on the sample names would also help – for example, including the blind sample morphological ID or given market name in the sample name.

Suggested grammatical changes:
L20: Change “comprises of” to “is comprised of”.
L38-39: Change “as well as monitoring” to “and monitoring”.
L41: Change “and the complex supply networks” to “and complex supply networks”.
L66: Change “is deposited” to “are deposited”.
L89: Change “respectively, 2)” to “respectively, and 2)”.
L91: Change “diversified” to “diverged”.
L136-137: Change “as well as shallow” to “and shallow”.
L157: Change “among four European” to “in four northern European”.
L182: Change “Both mitochondrial” to “The two mitochondrial”.
L226: Change “affected by” to “compromised by”.
L250: Change “and/or lacks the diagnostic polymorphisms to accurately identify them.” to “and/or a lack of diagnostic polymorphisms for identification.”
L265: Change “a good source of reference” to “a reliable reference”.
L270: Change “, in those groups known to hybridize” to “in hybridizing groups”.
L285: Change “by visual eye” to “visually”.
L341: Change “amounts” to “numbers”.

---

## Round 0.2 · accepted · Accept

Thank you for making the rebuttal letter and corrections to the manuscript so clear. I'm happy to accept the manuscript for publication in PeerJ